# Exposure to the Danish Mandatory Vitamin D Fortification Policy in Prenatal Life and the Risk of Developing Coeliac Disease—The Importance of Season: A Semi Ecological Study

**DOI:** 10.3390/nu12051243

**Published:** 2020-04-27

**Authors:** Caroline Moos, Katrine S. Duus, Peder Frederiksen, Berit L. Heitmann, Vibeke Andersen

**Affiliations:** 1Focused Research Unit for Molecular Diagnostic and Clinical Research, Institute of Regional Health Research, University Hospital of Southern Denmark, 6200 Aabenraa, Denmark; katrinesideniusduus@gmail.com (K.S.D.); va@rsyd.dk (V.A.); 2Research Unit for Dietary Studies at The Parker Institute, Bispebjerg and Frederiksberg Hospital, Part of the Copenhagen University Hospital, 2000 Frederiksberg, Denmark; eek@garp.dk (P.F.); Berit.lilienthal.heitmann@regionh.dk (B.L.H.); 3The Department of Public Health, Section for General Practice, University of Copenhagen, 1017 Copenhagen, Denmark; 4Institute of Molecular Medicine, University of Southern Denmark, 5230 Odense, Denmark

**Keywords:** vitamin D, coeliac disease, fetal programming, season of birth, prenatal exposure, fortification, D-tect, social experiment, public health epidemiology, ecological study

## Abstract

Few studies have examined the role of maternal diet in relation to development of coeliac disease (CD). In Denmark, cancellation of mandatory vitamin D fortification of margarine in June 1985 provided this opportunity. This study examined if season of birth or prenatal exposure to extra vitamin D from food fortification were associated with developing CD later in life. A strength of this study is the distinctly longer follow-up of patients (30 years). This register-based study has a semi-ecological design. Logistic regression analysis was used to estimate odds ratios and to calculate 95% confidence intervals. The odds ratio for developing CD was 0.81 (95% CI 0.66; 1.00 *p* = 0.054), comparing those with fetal exposure to mandatory vitamin D fortification policy of margarine to those without after adjusting for gender and season of birth. There was a statistically significant season effect particularly for children born in autumn (OR 1.6 95% CI 1.16; 2.21) and born in summer (OR 1.5 95% CI 1.1; 2.1) when compared to children born in winter. Although this study did not find evidence to support the premise that prenatal exposure to small extra amounts of vitamin D from a mandatory food fortification policy lowered risk of developing CD, the small number of CD cases and observed association between season of birth and CD suggest that environmental exposure ought to be further explored.

## 1. Introduction

There has been an alarming increase over the past 30 years in the prevalence and incidence of autoimmune diseases, e.g., multiple sclerosis, type 1 diabetes, Crohn’s disease, Graves’ disease, and coeliac disease (CD) [1]. In Denmark, incidence rates for CD have risen from 1.6 per 100,000 person-years in 1980–1984 to 15.2 per 100,000 person-years in 2015–2016 [2]. Previous prevalence of CD from the Danish registers has been reported as 50 per 100,000 [3]. However, more recent studies have estimated a higher prevalence of between 180 per 100,000 [2] or approximately 500 per 100,000 if screening and clinical evaluation were part of case identification [3]. Advice for primary prevention of CD includes breastfeeding and introduction of small amounts of gluten whilst breastfeeding although studies have not shown that this advice has long-term preventative effects [4,5] The only treatment available for these patients is an inconvenient and often expensive gluten-free diet [4,6] consisting of avoiding products containing wheat, barley, oats, and rye.

CD is a particularly unique autoimmune disease because it is well known that gluten (a protein found in wheat, barley, and rye) drives the dysfunction of the immune system in genetically susceptible individuals [7]. The reason why some people react to gluten and others do not may be a combination of genetic predisposition or environmental triggers [4].

The genetics of CD are well established with CD occurring almost exclusively with genotype HLA (human leucocyte antigens)-DQ2 or HLA-DQ8 haplotypes. The HLA-DQ2 haplotype has been reported in 90% of patients diagnosed with CD, and the HLA-DQ8 haplotype has been reported in 5% of patients. The remaining 5% of patients have at least one of the two genes encoding DQ2 [6]. Despite, 25–40% of the Caucasian population having these genes, only 2–3% develop CD after intake of gluten [5,8]. Therefore, the precise triggers for developing CD remain elusive. To date, a variety of possible triggers have been explored including (i) breastfeeding and infant feeding practices such as age at gluten exposure and ongoing breastfeeding at gluten exposure, (ii) pregnancy and birth-related factors including maternal antibiotic use and mode of delivery, (iii) maternal smoking (throughout pregnancy or never), and (iv) socioeconomic status [4,5,9]. All these factors have been explored using different types of studies, however, often with conflicting results.

More recently, maternal vitamin D levels have been suggested to have a role in the development of autoimmune diseases [10,11]. Other studies have shown an association between season of birth or latitude of place of living and CD [12,13,14,15,16,17]. These observations have led to a proposition known as the vitamin D deficiency–CD hypothesis [18]. This hypothesis suggests that maternal vitamin D deficiency dysregulates a neonate’s immune response in genetically predisposed individuals, resulting in the development of CD.

Vitamin D has been found to play an important role in promoting immunity by enhancing the innate immune system and by regulating the adaptive immune system [10,19]. A 2018 randomized control trial found a stronger innate cytokine response in neonates born from mothers supplemented from the second trimester with 4400IU/d vitamin D compared to mothers supplemented with 400IU/d [20]. However, two studies in 2017 did not find an association between prenatal vitamin D levels and the risk of CD [21,22]. The TEDDY study was an international multicenter observational study prospectively following 8676 children from birth until 15 years of age for environmental factors involved in both type 1 diabetes and CD. Its strengths included a large study population and the prospective multi-center study design, and weaknesses include the retrospective collection of information on diet intake during pregnancy and, therefore, the high likelihood of recall bias. The other study was a Norwegian nested case-control study using the Norwegian Mother and Child Cohort (MoBa), which consisted of 113,053 children. This study assessed 416 children with CD and 570 mother and child pairs as controls selected randomly from the MoBa cohort. The strengths of this study included repeated assessment of vitamin D levels, the prospective collection of information on vitamin D intake, and the inclusion of genetic risk factors. The limitations included possible selection bias, lack of information on sunlight levels, and age of the population in the cohort being relatively young (CD cases identified up to 7–8 years of age).

On 01 June 1985, the Danish Government abandoned the mandatory fortification program of adding vitamin D to margarine that had been in place since the mid-1930s. This change in policy gives a unique opportunity for studying the effect of vitamin D fortification. One such opportunity is the D-tect research program. This program was initiated to explore if the individual risk of disease differs between exposed (born during fortification of margarine) or unexposed offspring (born after fortification of margarine ceased) [23]. If the vitamin D fortification policy had a protective effect for the neonate in relation to later development of disease such as CD, it has immediate, cheap, and easy public health implications and mandatory food fortification in Denmark ought to be reconsidered.

The objectives of this study were as follows:(i)To investigate if season of birth and the concomitant production of vitamin D from sunlight was associated with the risk of developing CD(ii)To investigate if individuals with fetal exposure to extra vitamin D from the mandatory vitamin D food fortification policy had a decreased risk of developing CD later in life compared to individuals with no fetal exposure(iii)To examine if the risk of developing CD related to prenatal exposure to extra vitamin D from the fortification policy was dependent on season of birth and if risk reduction is potentially stronger for summer born compared to winter born children

## 2. Materials and Methods

### 2.1. Study Design

This was a semi-ecological study [24]. The study examined two cohorts exposed or not exposed to extra vitamin D from the Danish mandatory fortification policy. This policy stipulated that margarine was fortified with 1.25 µg vitamin D per 100g margarine. The exposure (maternal exposure to the fortification policy) is aggregated, exploratory, and global with no distinct measurement on an individual level. In contrast, the outcome, CD, was identified individually. The date of cancellation of this policy, 01 June 1985, serves as a time reference point for the creation of the two cohorts [25].

### 2.2. Study Population

The exposed study population consisted of all children born in Denmark in the two-year period between 01 June 1983 and 31 May 1985 (these children were exposed to the vitamin D fortification policy prenatally) and all children in Denmark born in the two year period between 01 September 1986 and 31 August 1988 (these children were not exposed to vitamin D fortification policy prenatally). Individuals who died or emigrated were excluded. See the flowchart of the study population (Figure 1). The cohorts were created with a washout period of 15 months, and records of all individuals in the study population were reviewed over a 30-year period for development of CD.

### 2.3. Washout Period

The 15-month washout period between the two cohorts consisted of 4 months to account for normal margarine shelf-life, an additional 2 months to account for margarine stored beyond shelf life, and a further 9 months to exclude individuals conceived during this 6-month period.

### 2.4. Sources of Data

The Danish National Patient register (DNPR) was established in 1977 [26,27] and was used to retrieve information on patients diagnosed with CD using the International Classification of Diseases (ICD-8 codes 269 until the end of 1993 and thereafter ICD-10 DK900). The Danish Civil Registration System (CPR) was established in 1968 and assigns a unique 10 digit number to everyone living in Denmark [28]. This number can be linked to other registries such as DNPR, enabling the study of large populations [29]. The CPR was used to retrieve information such as death, emigration, gender, and date of birth of individuals in the entire study population.

### 2.5. Exposure(s)

Season of birth is a categorical variable classified as follows: winter (November, December, and January), spring (February, March, and April), summer (May, June, and July), and autumn (August, September, and October). This categorization is based upon the seasonal fluctuation of vitamin D levels in individuals living in high northern latitude countries [30] and is consistent with classification of seasons from previous D-tect studies based on the same study population.

The abandonment of the mandatory vitamin D fortification policy marks the time point that separates the cohorts and is a binary categorical variable.

### 2.6. Outcome—Coeliac Disease

The outcome, CD diagnosis, is a binary categorical variable. To reduce the risk of false positives, we defined CD a priori as the occurrence of at least two or more records in the DNPR of ICD-8 code 269 and/or ICD-10 code K900. A previous study reported that the positive predictive value of identifying CD cases from DNPR if at least two records were used was 73.6–85.6% [31]. For the purposes of this study, diagnosis of dermatitis herpetiformis, a cutaneous manifestation of CD assisted by a gluten free diet [32], was not included as a CD outcome.

### 2.7. Statistical Analysis

A power calculation determined that, with an expected 350 cases of CD among the approximately 222,000 individuals in our study, with significance level of 5%, we have 80% power to detect an odds ratio of 0.75 from the exposed to the unexposed.

Logistic regression analyses were used to estimate odds ratios and to compute 95% confidence intervals. Analyses were adjusted for season of birth and sex. A multiplicative interaction between season of birth and exposure to vitamin D policy was tested by a likelihood ratio test using a 5% significance level. Sensitivity analyses were completed by changing the case definition to at least one record of CD in the DNPR and recategorization of seasons to winter (December, January, and February), spring (March, April, and May), summer (June, July, and August), and autumn (September, October, and November). All statistical analyses were performed using R version 3.3.3 (R Foundation for Statistical Computing, Vienna, Austria, www.R-project.org).

### 2.8. Ethics

According to Danish law, ethical approval is not required for register-based studies. Permission to access data was granted by the Danish Health Data Authority. The Centre of excellence in research in the Capital Region of Denmark provided permission to process the data (VD-2019-237).

## 3. Results

The total number of children available was 217,249 of which 206,900 were included in the study; 98,856 children were exposed prenatally to the fortification policy, and 108,044 children were not exposed. Of the exposed children, 148 (0.15%) developed CD during the subsequent 30 years, compared to 199 (0.18%) of unexposed children (Figure 1). There was a higher number of women than men who developed CD, with the majority being diagnosed after the age of 15 years (Table 1).

Season of birth was significantly associated with a change in the odds of developing CD, particularly for subjects born in autumn (OR = 1.60, 95% CI 1.16; 2.21 *p* = 0.004) and summer (OR = 1.51, 95% CI 1.10; 2.09 *p* = 0.01) compared to subjects born in winter (Table 2).

The odds ratio for developing CD was 0.81 (95% CI 0.66; 1.00 *p* = 0.054), comparing those with fetal exposure to mandatory vitamin D fortification policy of margarine to those without after adjusting for gender and season of birth (Table 2).

There was no significant interaction between season of birth and the prenatal exposure to the mandatory vitamin D fortification policy on the odds of developing CD later in life (*p* = 0.56).

## 4. Discussion

This is the first study to report a significant association between season of birth and risk of developing CD in a Danish population. This is also the first study to report on the association between prenatal exposure to extra vitamin D from a food fortification policy and the development of CD before 30 years of age.

There was a significant seasonal effect for individuals born in summer and autumn. However, while the overall results suggested a decreased risk of developing CD of almost 20% for those exposed prenatally to the extra vitamin D from mandatory fortification, the association was only borderline significant possibly due to the small number of cases with CD. There was no further benefit of the extra vitamin D for those born in summer or autumn seasons, where extra vitamin D during the early “dark” trimesters was hypothesized to have had a benefit.

### 4.1. Season of Birth and Risk of Developing CD

In this study, subjects born in winter had the lowest odds of developing CD. Inadequate exposure to sunlight is a major risk factor for vitamin D deficiency [33], and from October through April, where the ultraviolet B (UVB) radiation is insufficient for vitamin D conversion in the skin [34,35], Danes have lower vitamin D status and are predisposed to vitamin D deficiency [34]. Indeed, other immune diseases such as multiple sclerosis have been reported to have a higher prevalence in countries of high latitude [33]. CD was also reported as more common among people living at higher latitudes in the US despite adjustment for BMI and demographic factors [17]. It could be reasonably assumed that children born in summer or autumn in countries of high latitude have been exposed to lower maternal vitamin D levels in early to mid-gestation due to darker seasonal months in the first half of pregnancy. This may adversely influence the development of the fetus’ immune system with consequences later in life [36]. In agreement, earlier studies have reported a lower risk of developing CD for winter-born compared to summer-born children [12,13,14,15,16,37,38].

Although vitamin D variations provide a possible explanation for the association between CD risk and season of birth, the timing of weaning and first exposure to gluten in relation to viral load could also be responsible for the association with season. Indeed, children born in summer/autumn often begin weaning and have their first exposure to gluten during the winter period, which is characterized by higher exposure to seasonal viruses. Viral infections influence flora and permeability of the intestine, and other studies have suggested that this may play a role in the etiology of CD [16,39]. Indeed, a recent study reported that reovirus infection may trigger CD in some patients [40]. Thus, the development of viral infections at the time of introduction to gluten may also provide a further or additional explanation for the association between season of birth and CD.

### 4.2. Exposure to Extra Vitamin D from Fortification and The Development of CD

Although in the present study the association between prenatal exposure to the extra vitamin D from mandatory margarine fortification and the development of CD was not significant, the almost 20% reduced odds of CD indicate that there may be a protective effect of vitamin D fortification. Arguably, both the small extra amount of vitamin D from fortification, which contributed on average 13% of total dietary vitamin D [41], as well as the small number of CD cases in this cohort may have further contributed to the uncertainty of the fortification effect based on the broad confidence interval. On the other hand, the inconclusive association found in the present study is supported by findings from two previous studies that examined the association between prenatal exposure to vitamin D and development of CD [21,22]. In the case-control study from the Norwegian Mother and Child Cohort Study (MoBa), the authors reported that there was no association between maternal and neonatal vitamin D levels and CD [21]. In the longitudinal prospective TEDDY study, maternal use of vitamin D supplements during pregnancy did not influence the risk of the offspring developing CD [22].

However, our study had a distinctly longer follow-up (30 years vs 6–8 years) compared to TEDDY and MoBa [21,22], and although the onset of CD can occur throughout all ages of life, there are two reported peaks: one in the first 2 years of life and the other after the age of 20 or 30 [42]. Indeed, in our study, most individuals were diagnosed with CD after 15 years of age (See Table 1) making a direct comparison with the TEDDY and MoBa study results difficult.

It is difficult to pinpoint an exact timeframe during gestation where vitamin D deficiency may have the most critical effect on the developing fetus and its immune system. Recent discoveries have found that the fetal immune system is not simply immature or less responsive than the adult immune system but has its own specific and unique function [43]. Dendritic cells, crucial for a well-functioning immune system, have been reported to be fully functional at 13 weeks gestation but unique as they activate T-cells, suppressing the immune responses to foreign proteins rather than targeting these foreign proteins for destruction [43]. Vitamin D does have both immunoregulatory and anti-inflammatory effects on dendritic cells so the critical timeframe could be as early as 13 weeks. Another study suggested that the period was even earlier and may well be the irreversible critical period in particular in relation to gene transcription in the placenta and during placentation [44]. A recent randomized control trial exploring the programming of the immune system during fetal development reported that vitamin D supplementation beginning in the second trimester did modify the neonates immune system [20]. Much of the previous literature reports that vitamin D can affect the immune system, but our understanding of how this occurs is only just beginning to evolve [44]. Until development of the fetal immune system and the modification of genes and cells by nutrients is better understood, it will be difficult to ascertain accurately whether and when is the most desirable timing of vitamin D supplementation during pregnancy and the specific disease outcomes that timely vitamin D supplementation could alleviate.

### 4.3. Strengths and Limitations

A major strength of this study is the use of valid and complete register data and complete birth cohorts, ensuring that the results are generalizable to the entire Danish population. The unique long follow-up and clear definition of cases (minimizing false positives) are other strengths of the study design. No major societal events have been reported between June 1983 and August 1988 other than the termination of the vitamin D fortification policy [45,46]. Therefore, adjustment for confounding is unnecessary as the potential confounding factors are likely to be equally distributed between these two cohorts. However, residual confounding could occur if a secular trend is overlooked. Some of these potential trends are as follows:

Sources of vitamin D: The possibility that the two cohorts had different sources of vitamin D was explored as a potential confounder. However, vitamin D supplementation recommendations for pregnant women did not change during 1983–1988 [46]. The differences in monthly sunshine hours per year between 1983–1988 were not significant [47,48]. The consumption of margarine during this period was remarkably stable [49]. As a result of the above, the assumption has been made that sources of vitamin D were not different between the two cohorts.

Trends in CD incidence: CD has increased from 1.6 per 100,000 person years to 15.2 per 100,000 person years from 1986 to 2016 in Denmark [2]. This increased incidence in CD may well be the reason for the higher number of CD cases in unexposed cohort. There is no great difference in the number of CD cases between the two cohorts despite this increasing incidence, and therefore, the conclusion in relation to the lack of association between fortification and risk of CD is strengthened.

Major societal incident: The nuclear accident (Chernobyl) that occurred 26 April 1986 (during washout period) in the north of Ukraine may have increased the disease burden of the unexposed cohort. Denmark was one of the first countries to receive fallout from Chernobyl, but contamination was relatively modest due to the absence of precipitation when the first cloud passed the country [50]. This incident also strengthens our conclusion that exposure prenatally to fortification had no effect on risk of developing CD as the disease burden from such an accident would be expected to increase the strength of the association.

Apart from these secular trends, another significant limitation was that individual vitamin D levels were not assessed. This is a limitation of a register study with an aggregated exposure that cannot be measured on an individual level. Other limitations include the fact that the number of true cases in this study were most likely underestimated and that there were potential unaccounted false negative cases. In fact, despite this study’s long follow-up period of 30 years, the individuals who developed CD in their late 20s would not necessarily have had the two records required to be defined as a CD case before the age of 30 and, therefore, may not have been identified in the study. Therefore, despite following individuals for 30 years, the follow-up period may still have been a limiting factor. Prevalence of CD is higher by a factor of 10 if diagnosed via screening and clinical detection compared to prevalence based on register-based data [3], indicating that CD is mainly treated in the primary care system and/or is underdiagnosed in Denmark. We therefore performed a sensitivity analysis where the case definition was redefined to individuals who had a minimum of one record of CD in the DNPR. However, despite this change, there was still no significant association between prenatal exposure to the fortification policy and risk of developing CD (see Appendix A). This supported the finding of no association between prenatal exposure to mandatory vitamin D fortification policy and risk of developing CD. Studies from D-tect define seasons according to sunlight variation, while some other seasonal studies use a slightly different definition. Therefore, a sensitivity analysis was completed with re-categorized seasons (see Appendix A).

## 5. Conclusions

Although this study did not find evidence to support the premise that prenatal exposure to small extra amounts of vitamin D from a mandatory food fortification policy lowered risk of developing CD, the small number of CD and observed association between season of birth and CD suggest that environmental exposure ought to be further explored.

## Figures and Tables

**Figure 1 nutrients-12-01243-f001:**
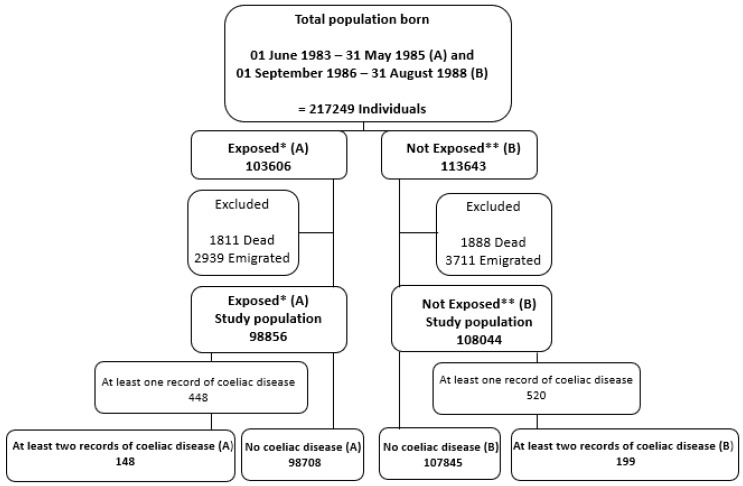
Flow chart of study population. * Exposed: The Danish population was exposed to the Danish vitamin D fortification policy. ** Not Exposed: The Danish population was not exposed to the Danish vitamin D fortification policy.

**Table 1 nutrients-12-01243-t001:** Characteristics of population with coeliac disease (CD) compared to those without (*N* = 206,900).

		CD (*n* = 347)		No CD (*n* = 206,553)		
		*n*	%	*n*	%	*p*–Value ^1^
Gender						<0.001
	Women	221	63.7	100,801	48.8	
	Men	126	36.3	105,752	51.2	
Season of birth						0.02
	Nov–Jan (winter)	58	16.7	46,696	22.6	
	Feb–Apr (spring)	83	23.9	52,745	25.5	
	May–Jul (summer)	103	29.7	55,064	26.7	
	Aug–Oct (autumn)	103	29.7	52,048	25.2	
Age at diagnosis ^2^						<0.001
	<2 years	100	28.8			
	2–14 years	54	15.6			
	15+ years	193	55.6			

^1^ Tested using Chi squared test. ^2^ Based on first record in the Danish National Patient register (DNPR).

**Table 2 nutrients-12-01243-t002:** Risk of coeliac disease among those prenatally exposed to extra vitamin D and dependent on season of birth.

		Odds Ratio	95% CI ^1^	*p*–Value
Vitamin D fortification policy ^2^				0.054
	Not exposed (ref)	1		
	exposed	0.81	0.66; 1.00	
Gender				<0.001
	Women (ref)	1		
	Men	0.54	0.44; 0.68	
Season of birth ^3^				0.02
	Nov–Jan (winter)(ref)	1		
	Feb–Apr (spring)	1.28	0.91; 1.78	0.16
	May–Jul (summer)	1.51	1.10; 2.09	0.01
	Aug–Oct (autumn)	1.60	1.16; 2.21	0.004

^1^ CI = Confidence interval. ^2^ Adjusted for sex and season of birth. ^3^ Adjusted for sex and the vitamin D fortification policy.

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
