# Peer review of "Exposure to the Danish Mandatory Vitamin D Fortification Policy in Prenatal Life and the Risk of Developing Coeliac Disease—The Importance of Season: A Semi Ecological Study"

_nutrients, 2020, doi:10.3390/nu12051243_

Round 1
Reviewer 1 Report
March 14, 2020
Manuscript ID: nutrients-748576
Type of manuscript: Article
Title: Exposure to the Danish mandatory vitamin D fortification policy in prenatal life and the risk of developing coeliac disease – the importance of season: A semi ecological study
Opinion:
This article is giving important information that there is no association between prenatal exposure to mandatory vitamin D fortification policy and the risk of developing CD. The result of this study is significant because this study had a distinctly longer follow-up of patients (30 years). However, the presentation of the study is weak. The abstract is not summarizing the significant information of the study. On some occasions, It seems that the authors are discussing a study that already has been already done. The authors should work on these. I have only some minor comments that I am summarizing below.
Comments to authors:
Minor comments:
Comment#1(Line no. 36): Authors should give the overall prevalence of CD.
Comment#2: Advice for prevention…. Which kind of prevention authors are discussing here? (Nine no. 37)
Comment#3 (Line no 39-40): The authors may mention that in CD the only treatment for CD i.e gluten-free diet, is recommended for “life-long”. They ideally should mention what is the major source of gluten (wheat, rye, etc).
Comment#4 (Line no 45-46): I request authors to recheck, In my view, about 100% of European CD patients display HLA-DQ2/8 positivity. Generally, 90-95% HLA-DQ2 and rest 5-10% HLA-D8
Comment#5: There are some minor grammatical mistakes that I hope the authors shall correct. Eg. Line no 45; “Genetics of CD are” genetics of CD is would be appropriate.
Comment#6: Figure 1: authors should clear the kind of exposure in boxes (Exposed, not exposed)
.
Author Response
Response to Reviewer's Comments
Overall comments Point 1: ‘However, the presentation of the study is weak. The abstract is not summarizing the significant information of the study’.
Response to overall comments from point 1: We have rewritten the abstract to take into account and highlighted the significant information
We have now changed the abstract from
Abstract: Few studies have examined the role of maternal diet in relation to development of coeliac disease (CD). In Denmark, cancellation of mandatory vitamin D fortification of margarine in June 1985 provided this opportunity. This study examined if season of birth or prenatal exposure to extra vitamin D from food fortification were associated with developing CD later in life. This register-based study has a partially ecologic design. Logistic regression analysis was used to estimate odds ratios and calculate 95% confidence intervals. There was a statistically significant season effect particularly for children born in autumn (OR 1.6 95% CI 1.16; 2.21) and born in summer (OR 1.5 95% CI 1.1;2.1) when compared to children born in winter. There was a lower but statistically insignificant odds ratio of developing CD (OR: 0.81 95% CI 0.65; 1.00) from the exposed birth cohort adjusted for sex and season of birth.
To:
Few studies have examined the role of maternal diet in relation to development of coeliac disease (CD). In Denmark, cancellation of mandatory vitamin D fortification of margarine in June 1985 provided this opportunity. This study examined if season of birth or prenatal exposure to extra vitamin D from food fortification were associated with developing CD later in life. A strength of this study is the distinctly longer follow-up of patients (30 years). This register-based study has a partially ecologic design. Logistic regression analysis was used to estimate odds ratios and calculate 95% confidence intervals. The odds ratio for developing CD was 0.81, (95%CI 0.66;1.00 p=0.054) comparing those with fetal exposure to mandatory vitamin D fortification policy of margarine to those without, and after adjusting for gender and season of birth. There was a statistically significant season effect particularly for children born in autumn (OR 1.6 95% CI 1.16; 2.21) and born in summer (OR 1.5 95% CI 1.1;2.1) when compared to children born in winter. Although this study did not find evidence to support the premise that prenatal exposure to small extra amounts of vitamin D from a mandatory food fortification policy lowered risk of developing CD, the small number of CD and observed association between season of birth and CD suggests that environmental exposure ought to be further explored
Overall comments point 2: ‘On some occasions, It seems that the authors are discussing a study that already has been already done.
Response to overall comments point 2: The tense and writing from passive to active has been changed to address this issue.
We have changed Line #104 from:
This change in policy gave researchers a unique opportunity. The D-tect research program was initiated aiming to explore if the individual risk of disease differs for exposed offspring (born during fortification of margarine) or unexposed (born after fortification of margarine ceased) [23].
To
This change in policy gives a unique opportunity for studying the effect of vitamin D fortification. One such study is the D-tect research program. This program was initiated aiming to explore if the individual risk of disease differs for exposed offspring (born during fortification of margarine) or unexposed (born after fortification of margarine ceased) [23].
We have changed Line #158 from
To reduce the risk of false positives, CD was defined a priori as the occurrence of at least two or more records in the DNPR of ICD-8 code 269 and/or ICD-10 code K900
To
To reduce the risk of false positives, we defined CD a priori as the occurrence of at least two or more records in the DNPR of ICD-8 code 269 and/or ICD-10 code K900.
Comment#1(Line no. 36): Authors should give the overall prevalence of CD.
Response#1: Yes, we agree the prevalence of CD needs to be better defined. We have now changed the text from
In Denmark, incidence rates for CD have risen from 1.6 per 100,000 person-years in 1980-1984 to 15.2 per 100,000 person-years in 2015-16 [2] and current prevalence is estimated to be 50 per 100,000 people [3].
To…
In Denmark, incidence rates for CD have risen from 1.6 per 100,000 person-years in 1980-1984 to 15.2 per 100,000 person-years in 2015-16 [2]. Previous prevalence of CD from the Danish registers has been reported as 50 per 100,000 [3]. However more recent studies have estimated a higher prevalence of between 180 per 100,000 [2] or approximately 500 per 100,000 if screening and clinical evaluation was part of case identification [3].
Comment#2: Advice for prevention…. Which kind of prevention authors are discussing here? (Line no. 37)
Response#2 We have now altered the text as follows from
Advice for prevention includes breastfeeding and introduction of small amounts of gluten whilst breastfeeding although studies have not shown that this advice has longer-term preventative effects [4,5].
To…..
Advice for primary prevention of CD includes breastfeeding and introduction of small amounts of gluten whilst breastfeeding although studies have not shown that this advice has long- term preventative effects [4,5].
Comment#3 (Line no 39-40): The authors may mention that in CD the only treatment for CD i.e gluten-free diet, is recommended for “life-long”. They ideally should mention what is the major source of gluten (wheat, rye, etc).
Response#3 We have now altered the text from
The only treatment available for these patients is an inconvenient and often expensive gluten-free diet [4,6]
To…
The only treatment available for these patients is an inconvenient and often expensive gluten-free diet [4,6] consisting of avoiding products containing wheat, barley, oats and rye.
Comment#4 (Line no 45-46): I request authors to recheck, In my view, about 100% of European CD patients display HLA-DQ2/8 positivity. Generally, 90-95% HLA-DQ2 and rest 5-10% HLA-D8
Response#4 Thanks for alerting us to this error. We have found it well reported in MacAllisters paper from 2018. We have now changed the text from:
The genetics of CD are well reported with approximately 90% of all CD patients having genotype HLA DQ2/8 (human leucocyte antigens DQ2 and/or DQ8) [8].
To…..
The genetics of CD are well established with CD occurring almost exclusively with genotype HLA (human leucocyte antigens) -DQ2 or HLA -DQ8 haplotypes. The HLA-DQ2 haplotype has been reported in 90% of patients diagnosed with CD and the HLA-DQ8 haplotype in 5% of patients. The remaining 5% of patients have at least one of the two genes encoding DQ2 [6]
Comment#5: There are some minor grammatical mistakes that I hope the authors shall correct. Eg. Line no 45; “Genetics of CD are” genetics of CD is would be appropriate.
Response#5: We have identified the following grammatical errors and corrected them
- Line#45 Genetics to genetics
- Line#50 Graves to Graves’
- Line #160 ICD 8 and ICD 10 to ICD-8 and ICD-10
- Line #297 ´long- to long
- Line #328 follow up to follow-up
Comment#6: Figure 1: authors should clear the kind of exposure in boxes (Exposed, not exposed)
Response#6: We have now added ‘*’ to the word ‘Exposed and ‘**’ to the words ‘Not exposed’ in Figure 1 and then adding the following footnote
*Exposed: The Danish population was exposed to the Danish vitamin D fortification policy
**Not Exposed: The Danish population was not exposed to the Danish vitamin D fortification policy
Reviewer 2 Report
In this study, the effect of vitamin D fortification policy (in margarine for the neonate in relation to later development of coeliac disease (CD) was examined. Another aim was to investigate if season of birth was associated with the risk of developing CD. This study find that children who were born in the period with prenatal exposure to the extra vitamin 25 D had a lower, but non-significant, risk of CD as compared to those with no prenatal exposure to vitamin D fortification. Moreover, the authors showed that children who were born in the summer or autumn had a higher risk of CD.
This well-written study has well-operationalised PICO research questions that have been answered with an original design.
A limitation is that vitamin D was not assessed on an individual level.
Minor comments:
Introduction:
“However, two studies in 2017 did not find an association between prenatal vitamin D levels and the risk of CD [21,22].” Please can you give some more information of these studies, e.g. the type of design and number of patients of these studies, strengthts and limitations.
Statistical analysis:
Can you describe how the interaction term between exposure to vitamin D policy and season of birth was calculated? Was this a multiplicative (vitamin D policy * season of birth) interaction term?
Results
The authors wrote that there was no significant association (OR = 0.81, 95%CI 0.66;1.00 p=0.054) between fetal exposure to mandatory vitamin D fortification policy of margarine and the risk of developing CD later in life (Table 2).
Although the association was not significant, the OR indicates that there is a protective effect of vitamin D fortification, but the 95% Confidence interval is broad indicating that the magnitude of the effect estimation is uncertain. I would prefer to focus on the magnitude of the effect (OR) and the 95% CI rather than on the significance. Please can you change the results, discussion and conclusion accordingly.
Discussion
- After reference [34,35] there should be , instead of a .
- I would suggest to modify the statement “This study did not find a conclusive association between prenatal exposure to the extra vitamin 205 D from mandatory margarine…” into “This study did find a non-significant protective association between prenatal exposure to the extra vitamin 25 D …..”
- In the discussion, the authors can explain that the large uncertainty may be due to the small power (number of cases with CD).
- Please mention as limitation that vitamin D serum levels were not assessed.
Author Response
Response to Reviewer's Comments
Comment #1 Introduction: “However, two studies in 2017 did not find an association between prenatal vitamin D levels and the risk of CD [21,22].” Please can you give some more information of these studies, e.g. the type of design and number of patients of these studies, strengths and limitations.
Response#1 We have now added the following text to the sentence
“However, two studies in 2017 did not find an association between prenatal vitamin D levels and the risk of CD [21,22].” The TEDDY study was an international multicenter observational study prospectively following 8676 children from birth until 15 years for environmental factors involved in both type 1 diabetes and CD. Its strengths included a large study population and the prospective multi-center study design and weaknesses include the retrospective collection of information of diet intake during pregnancy, and therefore, the high likelihood of recall bias. The other study was a Norwegian nested case-control study using the Norwegian Mother and Child Cohort (MoBa), which consisted of 113,053 children. This study assessed 416 children with CD and 570 mother and child pairs as controls selected randomly from the MoBa cohort. The strengths of this study included repeated assessment of vitamin D levels, the prospective collection of information on vitamin D intake and the inclusion of genetic risk factors. The limitations included possible selection bias, lack of information on sunlight levels and age of the population in the cohort was relatively young (CD cases identified up to 7-8 years of age).
Comment #2 Statistical analysis :Can you describe how the interaction term between exposure to vitamin D policy and season of birth was calculated? Was this a multiplicative (vitamin D policy * season of birth) interaction term?
Response#2 To test for interaction the models were augmented with a product term. But since everything is estimated on a log-odds scale its actually an additive interaction. It translates into a multiplicative interaction on the odds scale.
We have now changed the text (Line 31) from
Effects of season of birth and interaction between exposure to vitamin D policy and season of birth were tested by a likelihood ratio tests using a 5 % significance level.
To
A multiplicative interaction between season of birth and exposure to vitamin D policy was tested by a likelihood ratio tests using a 5% significance level.
Comment #3 Results: The authors wrote that there was no significant association (OR = 0.81, 95%CI 0.66;1.00 p=0.054) between fetal exposure to mandatory vitamin D fortification policy of margarine and the risk of developing CD later in life (Table 2). Although the association was not significant, the OR indicates that there is a protective effect of vitamin D fortification, but the 95% Confidence interval is broad indicating that the magnitude of the effect estimation is uncertain. I would prefer to focus on the magnitude of the effect (OR) and the 95% CI rather than on the significance. Please can you change the results, discussion and conclusion accordingly.
Response#3 We have changed the text accordingly in the results, discussion, conclusion and abstract
Results (Line 155) changed from
There was no significant association (OR = 0.81, 95%CI 0.66;1.00 p=0.054) between fetal exposure to mandatory vitamin D fortification policy of margarine and the risk of developing CD later in life when adjusting for gender and season of birth (Table 2).
To…
The odds ratio for developing CD was 0.81, (95%CI 0.66;1.00 p=0.054) comparing those with fetal exposure to mandatory vitamin D fortification policy of margarine to those without, and after adjusting for gender and season of birth (Table 2).
Discussion (Line 177) changed from
However, the overall association between prenatal exposure to the extra vitamin D from mandatory fortification and the development of CD was not significant.
To….
However, while the overall results suggested a decreased risk developing CD of almost 20%for those exposed prenatally to the extra vitamin D from mandatory fortification the association was borderline significant only possibly due to the small number of cases with CD.
Discussion (Line 205) changed from
This study did not find a conclusive association between prenatal exposure to the extra vitamin D from mandatory margarine fortification and the development of CD. Arguably, the small extra amount of vitamin D from fortification, which contributed on average 13% of total dietary vitamin D [41], may have been too low to influence the risk of CD. This inconclusive association is supported by findings from two previous studies that examined the association between prenatal exposure to vitamin D and development of CD [21,22].
To:
Although in the present study, the association between prenatal exposure to the extra vitamin D from mandatory margarine fortification and the development of CD was not significant, the almost 20% reduced odds of CD indicates that there may be a protective effect of vitamin D fortification. Arguably, both the small extra amount of vitamin D from fortification, which contributed on average 13% of total dietary vitamin D [41], as well as the small number of CD cases in this cohort may have further contributed to the uncertainty of the fortification effect based on the broad confidence interval. On the other hand, the inconclusive association found in the present study is supported by findings from two previous studies that examined the association between prenatal exposure to vitamin D and development of CD [21,22].
Conclusion (Line 280) changed from
This study did not find evidence to support the premise that prenatal exposure to small extra amounts of vitamin D from a mandatory food fortification policy lowered risk of developing CD. The observed association between season of birth and CD suggests that environmental exposure with seasonal patterns ought to be further explored.
To…
Although this study did not find evidence to support the premise that prenatal exposure to small extra amounts of vitamin D from a mandatory food fortification policy lowered risk of developing CD, the small number of CD and observed association between season of birth and CD suggests that environmental exposure ought to be further explored.
Abstract changed from
Few studies have examined the role of maternal diet in relation to development of coeliac disease (CD). In Denmark, cancellation of mandatory vitamin D fortification of margarine in June 1985 provided this opportunity. This study examined if season of birth or prenatal exposure to extra vitamin D from food fortification were associated with developing CD later in life. This register-based study has a partially ecologic design. Logistic regression analysis was used to estimate odds ratios and calculate 95% confidence intervals. There was a statistically significant season effect particularly for children born in autumn (OR 1.6 95% CI 1.16; 2.21) and born in summer (OR 1.5 95% CI 1.1;2.1) when compared to children born in winter. There was a lower but statistically insignificant odds ratio of developing CD (OR: 0.81 95% CI 0.65; 1.00) from the exposed birth cohort adjusted for sex and season of birth.
To:
Few studies have examined the role of maternal diet in relation to development of coeliac disease (CD). In Denmark, cancellation of mandatory vitamin D fortification of margarine in June 1985 provided this opportunity. This study examined if season of birth or prenatal exposure to extra vitamin D from food fortification were associated with developing CD later in life. A strength of this study is the distinctly longer follow-up of patients (30 years). This register-based study has a partially ecologic design. Logistic regression analysis was used to estimate odds ratios and calculate 95% confidence intervals. The odds ratio for developing CD was 0.81, (95%CI 0.66;1.00 p=0.054) comparing those with fetal exposure to mandatory vitamin D fortification policy of margarine to those without, and after adjusting for gender and season of birth. There was a statistically significant season effect particularly for children born in autumn (OR 1.6 95% CI 1.16; 2.21) and born in summer (OR 1.5 95% CI 1.1;2.1) when compared to children born in winter. Although this study did not find evidence to support the premise that prenatal exposure to small extra amounts of vitamin D from a mandatory food fortification policy lowered risk of developing CD, the small number of CD and observed association between season of birth and CD suggests that environmental exposure ought to be further explored
Comment #4 Discussion
- After reference [34,35] there should be , instead of a .
- I would suggest to modify the statement “This study did not find a conclusive association between prenatal exposure to the extra vitamin 205 D from mandatory margarine…” into “This study did find a non-significant protective association between prenatal exposure to the extra vitamin 25 D …..”
- I would suggest to modify the statement “This study did not find a conclusive association between prenatal exposure to the extra vitamin 205 D from mandatory margarine…” into “This study did find a non-significant protective association between prenatal exposure to the extra vitamin 25 D …..”In the discussion, the authors can explain that the large uncertainty may be due to the small power (number of cases with CD).
- Please mention as limitation that vitamin D serum levels were not assessed.
Response#4
- We have now changed from
(Line 183) Inadequate exposure to sunlight is a major risk factor for vitamin D deficiency [33], and from October through April, where the ultraviolet B (UVB) radiation is insufficient for vitamin D conversion in the skin [34,35]. Danes have lower vitamin D status and are predisposed to vitamin D deficiency [34].
To:
Inadequate exposure to sunlight is a major risk factor for vitamin D deficiency [33], and from October through April, where the ultraviolet B (UVB) radiation is insufficient for vitamin D conversion in the skin [34,35], Danes have lower vitamin D status and are predisposed to vitamin D deficiency [34].
- We have now changed from:
(Line 205) This study did not find a conclusive association between prenatal exposure to the extra vitamin D from mandatory margarine fortification and the development of CD.
To:
This study did not find a conclusive association between prenatal exposure to the extra vitamin D from mandatory margarine fortification and the development of CD.
- We have now changed from:
Line 205: This study did not find a conclusive association between prenatal exposure to the extra vitamin D from mandatory margarine fortification and the development of CD. Arguably, the small extra amount of vitamin D from fortification, which contributed on average 13% of total dietary vitamin D [41], may have been too low to influence the risk of CD. This inconclusive association is supported by findings from two previous studies that examined the association between prenatal exposure to vitamin D and development of CD [21,22].
To:
Although in the present study, the association between prenatal exposure to the extra vitamin D from mandatory margarine fortification and the development of CD was not significant, the almost 20% reduced odds of CD indicates that there may be a protective effect of vitamin D fortification. Arguably, both the small extra amount of vitamin D from fortification, which contributed on average 13% of total dietary vitamin D [41], as well as the small number of CD cases in this cohort may have further contributed to the uncertainty of the fortification effect based on the broad confidence interval. On the other hand, the inconclusive association found in the present study is supported by findings from two previous studies that examined the association between prenatal exposure to vitamin D and development of CD [21,22].
- We have now changed from
Line 265 : Apart from these secular trends another potential limitation may be related to the fact that the number of true cases in this study were most likely underestimated and that there were potential unaccounted false negative cases.
To:
Apart from these secular trends another significant limitation was that individual vitamin D levels were not assessed. This is a limitation of a register-study with an aggregated exposure that cannot be measured on an individual level. Other limitations include the fact that the number of true cases in this study were most likely underestimated and that there were potential unaccounted false negative cases.